# Measuring under-5 mortality and fertility through mobile phone surveys: an assessment of selection bias in 34 low-income and middle-income countries

David A Sánchez-Páez [1,2] Bruno Masquelier [2] Ashira Menashe-Oren [2] Ololade Julius Baruwa [2,3] Georges Reniers [4]

¹Department of Economics, Universidad de Valladolid, Valladolid, Spain
²Centre for Demographic Research, Université catholique de Louvain, Louvain-la-Neuve, Belgium
³Centre for Social Science Research, University of Cape Town, Cape Town, South Africa
⁴Population Studies Group, London School of Hygiene and Tropical Medicine, London, UK

**Correspondence to**
Dr David A Sánchez-Páez; david.sanchezpaez@uva.es

## ABSTRACT

**Objectives** This study aims to assess sample selection bias in mobile phone survey estimates of fertility and under-5 mortality.

**Design** With data from the Demographic and Health Surveys, we use logistic regressions to identify sociodemographic correlates of mobile phone ownership and access, and Poisson regressions to estimate the association between mobile phone ownership (or access) and fertility and under-5 mortality estimates. We evaluate the potential reasons why estimates by mobile phone ownership differ using a set of behavioural characteristics.

**Setting** 34 low-income and middle-income countries, mostly in sub-Saharan Africa.

**Participants** 534 536 women between the ages of 15 and 49.

**Outcome measures** Under-5 mortality rate (U5MR) and total fertility rate (TFR).

**Results** Mobile phone ownership ranges from 23.6% in Burundi to 96.7% in Armenia. The median TFR ratio and U5MR ratio between the non-owners and the owners of a mobile phone are 1.48 and 1.29, respectively. Fertility and mortality rates would be biased downwards if estimates are only based on women who own or have access to mobile phones. Estimates of U5MR can be adjusted through poststratification using age, educational level, area of residence, wealth and marital status as weights. However, estimates of TFR remain biased even after adjusting for these covariates. This difference is associated with behavioural factors (eg, contraceptive use) that are not captured by the poststratification variables, but for which there are also differences between mobile phone owners and non-owners.

**Conclusions** Mobile phone surveys need to collect data on sociodemographic background characteristics to be able to weight and adjust mortality estimates ex post facto. Fertility estimates from mobile phone surveys will be biased unless further research uncovers the mechanisms driving the bias.

## STRENGTHS AND LIMITATIONS OF THIS STUDY

⇒ Mobile phone ownership and access are associated with sociodemographic background attributes, which could bias demographic estimates derived from data collected through mobile phone surveys.

⇒ We established that fertility and under-5 mortality estimates derived from birth histories administered for women who own or have access to a mobile phone will be biased downwards.

⇒ Poststratification on sociodemographic background characteristics corrects the bias in under-5 mortality estimates, but adjusted fertility estimates remain lower than in a nationally representative sample including women who do not own or have access to a mobile phone.

⇒ This study evaluates selection bias associated with the limitation of the sampling frame in a mobile phone survey, but does not evaluate bias that may result from selective participation in a mobile phone survey.

⇒ Mobile phone ownership is increasing rapidly in many low-income and middle-income countries, but its distribution will remain uneven for some time, so sample selection bias should be considered when deriving demographic estimates from mobile phone surveys.

## INTRODUCTION

Use of mobile phones has spread widely across the world, including in low-income and middle-income countries (LMICs) where the technological infrastructure for landlines is often lacking.[1 2] This spread has allowed populations to communicate faster and effortlessly, to access the internet, and it has accelerated social and economic development.[3 4] Widespread and increasing coverage of mobile phones also provides an opportunity to conduct mobile phone surveys (MPS). Some benefits of MPS include cost efficiency, shorter fieldwork duration due to reduced time devoted to travel, improved ability to supervise data collection through the recording of interviews and automated storage of paradata and increased access

to populations that are harder to reach.[5 6] MPS have already covered some health-related topics, including family planning and sexual and reproductive health,[1 7] but few MPS have included questions to monitor trends in fertility and mortality.

In countries lacking comprehensive systems of civil registration and vital statistics, data on mortality and fertility are generally collected through face-to-face interviews in survey programmes such as the Demographic and Health Surveys (DHS) or Multiple Indicator Cluster Surveys. However, face-to-face interviews might not be the most cost-effective approach and they are not suitable in settings affected by conflicts, disasters or epidemic outbreaks. During the Ebola epidemic in Liberia, MPS were used to estimate mortality both from Ebola and other causes based on reports of recent household deaths.[8] During the COVID-19 pandemic, MPS were proposed as a viable way to assess excess mortality without posing additional risks to survey teams and respondents.[9 10] MPS, thus, represent a promising alternative to collect demographic data.

MPS have several disadvantages that require careful consideration, such as higher non-response rates and shortened interview duration—requiring changes to the questionnaires used in face-to-face interviews.[11] Moreover, MPS may not be representative, especially when based on random digit dialling, as compared with a survey using phone numbers previously collected in face-to-face surveys.[1] This is because inequalities in use of digital technology, known as a 'digital divide', persist. Mobile phone use is related to gender, socioeconomic status, technological proficiency and literacy. Women have been noted to be at a disadvantage in mobile phone use, although the more common mobile phones are, the smaller the gender gap is.[12–14] This gender divide in mobile phone use reflects multiple barriers women face, including lack of financial resources, lack of time due to care responsibilities or household chores, sociocultural restrictions and lower educational attainment.[15] Mobile phone users are typically also better educated[16] and younger.[17] Moreover, rural dwellers use mobile phones relatively less than urbanites,[18] although overall access remains high in rural areas as well.[19] Other factors, not related directly to an individual's characteristics, also determine selectivity in mobile phone users. For example, in remote locations and near borders, cellular network coverage may be weak, leaving populations living in such areas underrepresented in MPS.

At the aggregate level, use of mobile phone has been linked to sustainable development.[2] The higher the diffusion of mobile phones, the lower under-5 (U5M) and maternal mortality, the greater the use of contraception, and the smaller ideal family size.[2 20] This is in part because mobile phone use is a good proxy for socioeconomic development, but also because mobile phone use can improve access to health services, including reproductive health and antenatal care.[21–25] Such benefits could hinder obtaining a representative sample, even after postsampling weighting. For example, a study conducted in Burkina Faso found higher modern contraception use among respondents reached by an MPS (40%) compared with respondents in a face-to-face survey (26%).[7] This difference in modern contraceptive use persisted after poststratification. Similarly, LeFevre et al[26] explored variations in several reproductive maternal, newborn and child health indicators, such as antenatal and postnatal care, skilled attendance at birth, measles and tetanus vaccinations, using DHS in 15 countries. Even after controlling for a wide range of sociodemographic variables, child health and prenatal care was better when mothers had a mobile phone. A notable exception was the adoption of initial, exclusive and continuous breast feeding, which was less common among mobile phone users.

These previous studies suggest that fertility and mortality estimates derived from MPS are likely to be biased, but this issue has not been systematically evaluated. In this study, we leverage existing DHS data containing information on mobile phone use by women to assess the selection bias in U5M and fertility estimates. We expect that these estimates are biased downwards, since mortality and fertility tend to be higher in lower socioeconomic groups.[27 28] We assess mobile phone use by distinguishing between women owning a mobile phone (henceforth referred to as 'ownership') and those who can potentially be reached through another household member who owns a phone (henceforth referred to as 'access'). We first identify correlates of mobile phone ownership and access through logistic regressions and establish through Poisson regressions whether ownership and access are associated with variations in U5M and fertility estimates.

## DATA AND METHODS
### Data collection
We used all available DHS[29] that include information on mobile phone ownership and access. The question on mobile phone ownership (v169a) is included in the individual questionnaire, while the question on access to a mobile phone (hv243a) is in the household questionnaire. Access to mobile phone means that the woman either owns a mobile phone or lives in a household where the household head owns a mobile phone. Some countries have more than one survey with such data, so in these cases, we used the most recent one. In total, we used 34 DHS, mostly from sub-Saharan Africa (see online supplemental appendix table 1). These surveys were collected between 2015 and 2021. All women have complete information for all variables of interest included in this study.

The survey India 2019–2021 is excluded from the analysis as the information is missing for about 85% of respondents. In Papua New Guinea, 2016–2018 information is missing for 0.4% of respondents, in Indonesia 2017 for 0.1% and in Pakistan 2017–2018 for 0.01%. We removed the observations with missing data from these three samples.

## Data analysis

Using data from the birth histories administered for women aged 15–49, we computed total fertility rates (TFR) for the 36-month period preceding the survey and under-5 mortality rates (U5MR) for the 10-year period preceding the survey. We used Poisson regressions to estimate the association between mobile phone ownership and access and TFR and U5MR. For fertility, the outcome variable was the number of births, while for U5M, it was the number of deaths before age 5. We conducted two sets of regressions. In the first set—unadjusted estimates—we regressed the outcome variable on the ownership of a mobile phone (model 1) or access to a mobile phone in the household (model 2), including covariates for the mother's age group when estimating fertility, and the child's age when estimating mortality. In the second set—adjusted estimates—we added educational level, place of residence, wealth and marital status as independent variables to models 1 and 2. As for the categories of these independent variables, place of residence has two values: urban and rural; educational level takes three values: no education, primary education and secondary and higher education; wealth has two values: poor (wealth quintiles 1 and 2) and not poor (wealth quintiles 3, 4 and 5); and, marital status takes two values: not in union (never married, not living together, divorced or widowed) and in union (married or living together). We also included exposure (person-years calculated for women for fertility and for children for mortality) as an offset in all models. In addition, we computed raking weights to match the marginal distributions of the same sociodemographic variables in the general population and examine whether potential estimation biases could be addressed using poststratification.

Differences in U5MR and TFR could be due to differences in women's behaviour, which could be captured by mobile phone use, but not by sociodemographic variables. We proposed a series of logistic regressions to analyse the association between mobile phone ownership and access, and some variables connected to U5MR and TFR. For mortality, we used six outcome variables. First, we modelled the likelihood of a mother having at least four antenatal care visits among those who gave birth in the 5 years before each survey. Second, the likelihood of a newborn receiving at least two signal functions as part of postnatal care (among births that occurred in the last 24 months). Third, the likelihood of a child being exclusively breastfed for the first 6 months (among the most recent birth in the last 6 months). Fourth, the likelihood of a child receiving eight basic immunisations among children aged 12–35 months at the time of the survey. The eight basic immunisations are one dose of bacille Calmette-Guerin (BCG), three doses of Diphtheria-pertussis-tetanus (DPT), three doses of polio (excluding at birth) and one dose of measles. Fifth, we analysed the likelihood of skilled attendance at birth, which includes the presence of a doctor, nurse or nurse-midwife during delivery. All births in the 5-year period before the survey

are considered. The last outcome variable connected with U5M that was analysed was the prevalence of underweight among children aged 0–5 at the time of the survey. A child is considered as being underweight if the weight-for-age z-score is below 2 SD below the mean on the WHO Child Growth Standards. On the other hand, for fertility, we used four outcome variables. First, the use of modern contraceptives. Second, sexual activity. We considered a woman to be sexually active when she reported having had sexual intercourse in the last month or being pregnant. Lastly, we analysed duration of postpartum amenorrhoea and duration of postpartum abstinence. For the latter two variables, we used a linear model since duration is measured as the number of months. Similar to the estimates of under-5 mortality and fertility, unadjusted and adjusted models were estimated for all outcome variables.

We conducted this analysis using the programming language for statistical computing and graphics R.[30] We used the following R packages: rdhs[31] for downloading the DHS datasets; tidyverse[32] for data manipulation and graphics; demogsurv[33] for preparing person-period tables stratified by the independent variables; survey[34] and broom[35] for econometric estimations and visualisation; anesrake[36] for computing raking weights to match the marginal distributions of sociodemographic variables for testing for poststratification; MetBrewer[37] for colour palettes in our plots; and kableExtra[38] and knitr[39] for writing and editing the manuscript.

## Patient and public involvement

Patients and/or the public were not involved in the design, or conduct, or reporting, or dissemination plans of this research.

## RESULTS

Across the 34 surveys used, the percentage of women owning a mobile phone ranges from 23.6% in Burundi to 96.7% in Armenia. The percentage of women who have access to a mobile phone ranges from 55% in Burundi to 100% in the Maldives (see online supplemental appendix tabe 1). The median ratio of the TFR calculated among non-owners to the TFR calculated among mobile phone owners is 1.48. Fertility is always higher among non-owners, with the lowest ratio at 1.04 in Bangladesh and the highest at 1.93 in Haiti. In absolute terms, these differences correspond to more than one child per woman in 25 of the 34 surveys studied here. Similarly, U5M is higher when calculated among non-owners, with the exception of South Africa 2016 and Zambia 2018–2019. The median ratio of U5MR calculated from reports of non-owners to that calculated from mothers with mobile phones is 1.29, and it ranges from 0.85 in South Africa to 2.5 in Albania. Differences in TFR and U5MR remain when access to a mobile phone is considered, but they are slightly reduced.

A series of logistic regressions (see online supplemental appendix figures 1–5) conducted for each survey separately indicate that women with mobile phones differ

from non-owners by age group, educational level, area of residence, wealth status and marital status. Except for age, differences do not disappear when access to a mobile phone instead of ownership is considered. Compared with young adults aged 20–24, adolescents aged 15–19 are less likely to own a phone in all countries, although this is not true in 27 countries when we consider access rather than ownership. Age patterns of phone ownership vary by country for older women, but in most cases, differences are not statistically significant when access is considered.

Compared with women without education, women with primary education are significantly more likely to own a mobile phone in 30 surveys and women with secondary or higher education are significantly more likely to own a mobile phone in all surveys. The number of surveys with a significant difference decreases, respectively, to 28 surveys and 30 surveys when access is considered. Women living in urban areas are significantly more likely to own a mobile phone in 29 surveys and to have access in 19 surveys. ORs for women who own a mobile phone range from 0.94 (95% CI 0.74 to 1.19) in South Africa to 5.42 (95% CI 4.23 to 6.95) in Ethiopia, while for access, from 0.41 (95% CI 0.25 to 0.67) in South Africa to 4.2 (95% CI 3.12 to 5.67) in Sierra Leone.

In terms of household assets, the gaps are very large between women in the poorest households (quintiles 1–2) and those living in the most affluent households (quintiles 3–5). Unsurprisingly, women living in households with more amenities are between 1.17 (Maldives, 95% CI 0.83 to 1.66) times and 4.85 (Burundi, 95% CI 4.24 to 5.55) times more likely to own a mobile phone. As for access, the ORs range between 2.7 (Mali, 95% CI 2.03 to 3.58) and 24.37 (Indonesia, 95% CI 18.01 to 32.97). Women not in union are more likely to own a mobile phone than women in union in 19 countries, with ORs ranging from 0.56 (95% CI 0.48 to 0.64) in Nepal to 1.78 (95% CI 1.4 to 2.26) in the Philippines.

Figures 1 and 2 present the exponentiated coefficients of mobile phone ownership and access from Poisson regressions on U5M and fertility, respectively. Left panels of both figures show unadjusted estimates, while right panels contain adjusted estimates. Unadjusted coefficients suggest that in most countries, ownership is associated with lower odds of experiencing a birth or the death of a child under 5 years of age. When considering U5M estimates and mobile phone ownership, coefficients are statistically significant in all countries but nine (unadjusted estimates in figure 1). However, in most countries differences by ownership are no longer significant once we adjust for marital status, education, wealth and place of residence (right panel of figure 1). Women who own a mobile phone are less likely to lose a child in only four countries (Cameroon, Gambia, Nepal and Nigeria) after adjusting for sociodemographic characteristics. Overall, U5MR is not statistically associated with owning or having access to a mobile phone, after controlling for basic sociodemographic characteristics. U5MR estimates obtained using poststratification weights from mobile phone

owners are in line with these findings (see adjusted estimates in figure 3). Only in Nigeria is it not possible to reproduce actual U5MR using ex post weightings. Results for access are presented in online supplemental appendix figure 6.

Unadjusted coefficients from the Poisson regressions on fertility for mobile phone owners are also statistically significant in 30 countries (left panel of figure 2). However, in contrast to what we observed for U5MR, the negative effect of owning a phone persists for fertility in most countries even after controlling for covariates (adjusted estimates in figure 2). Ethiopia is the only survey where this association loses significance after controlling for sociodemographic characteristics. When access is considered, the effect is not significant in five additional countries (Burundi, Cameroon, Mali, Pakistan and Senegal). In most countries, ownership or access is associated with lower fertility, even after controlling for basic sociodemographic characteristics. The use of poststratification weights based on the marginal distribution of sociodemographic variables of mobile phone owners shows that it is possible to reproduce the observed TFR in a total of 22 countries (see the empty blue circles in figure 3). However, this also means that poststratification cannot provide unbiased fertility estimates in one-third of the countries included in our study. When access is considered (see online supplemental appendix figure 6), there are no statistical differences with the TFRs observed in the general population. As mobile phone access is very high, marginal distributions of the sociodemographic variables of women with access are very similar to that of the whole population so poststratification plays a small role in this case.

In figure 4, we present the estimates of the association between ownership and variables connected to fertility. Results for access to mobile phones are presented in online supplemental appendix figure 7. We observe statistically significant differences in use of modern contraceptives in 23 surveys. In most of them, owners are more likely to use contraceptives. After controlling for covariates, differences remain in 20 surveys. When access is considered, the number of surveys is reduced to 16. In contrast, mobile phone owners are less likely to be sexually active in 14 surveys, even after controlling for sociodemographic characteristics. However, differences disappear in most surveys when access is considered. Postpartum amenorrhoea periods are significantly shorter among owners in 19 surveys, although only in 8 surveys after including controls. In contrast, owners have significantly longer postpartum abstinence periods in 15 surveys, but only in 8 surveys after adjusting for controls.

Analyses of the association between ownership and variables connected to U5M are presented in figure 5. Results for access are in online supplemental appendix figure 8. Owners are more likely to have had at least four antenatal care visits in all but three surveys. Differences are not statistically significant in six surveys after controlling for covariates. Likewise, births born to mothers who own

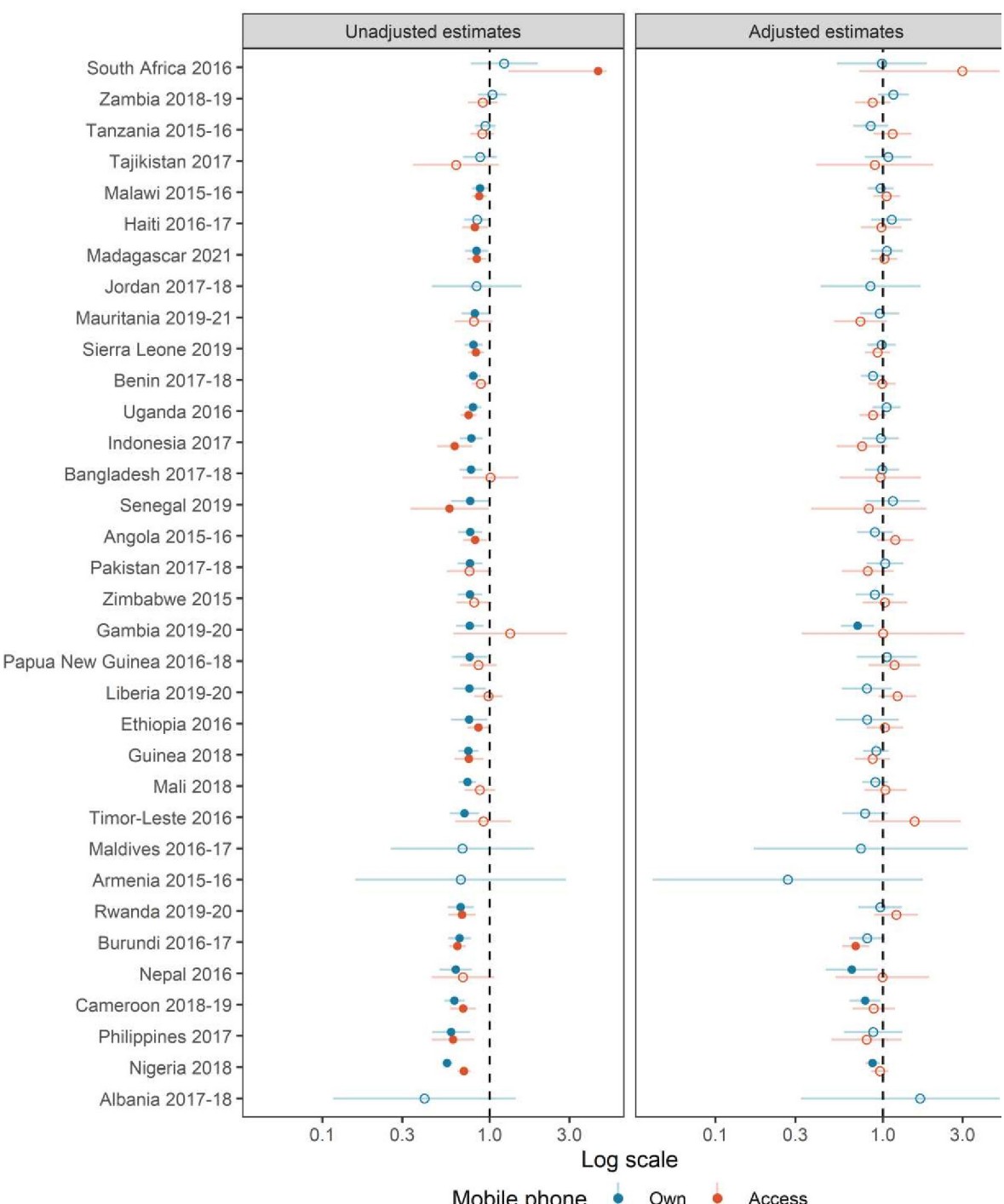

**Figure 1** Unadjusted and adjusted estimates of the effect of owning and having access to a mobile phone on under-5 mortality from Poisson regressions. Estimates for access to a mobile phone are not presented for Albania, Armenia, Jordan and Maldives, as access is almost 100%. Fill dots mean that estimates are statistically significant at the 95% confidence level, while empty dots mean that estimates are not statistically significant.

a mobile phone are more likely to have received at least two signal functions as part of postnatal check-ups in 28 surveys, although the number of surveys is reduced to 18 after controlling for covariates. In contrast, owners are more likely to exclusively breastfeed for the first 6 months to newborns in four surveys, and less likely in four other surveys, although these differences disappear in virtually all surveys after controlling for covariates. Children born to

owners are more likely to have received basic immunisation in only five surveys after adjusting for controls. In contrast, children born to owners are more likely to have been attended by health professionals at delivery in 27 surveys. Owners are less likely to have underweight children in more than half of the surveys, even after controlling for covariates. Differences disappear for most variables when access is considered. Exceptions are antenatal care, the

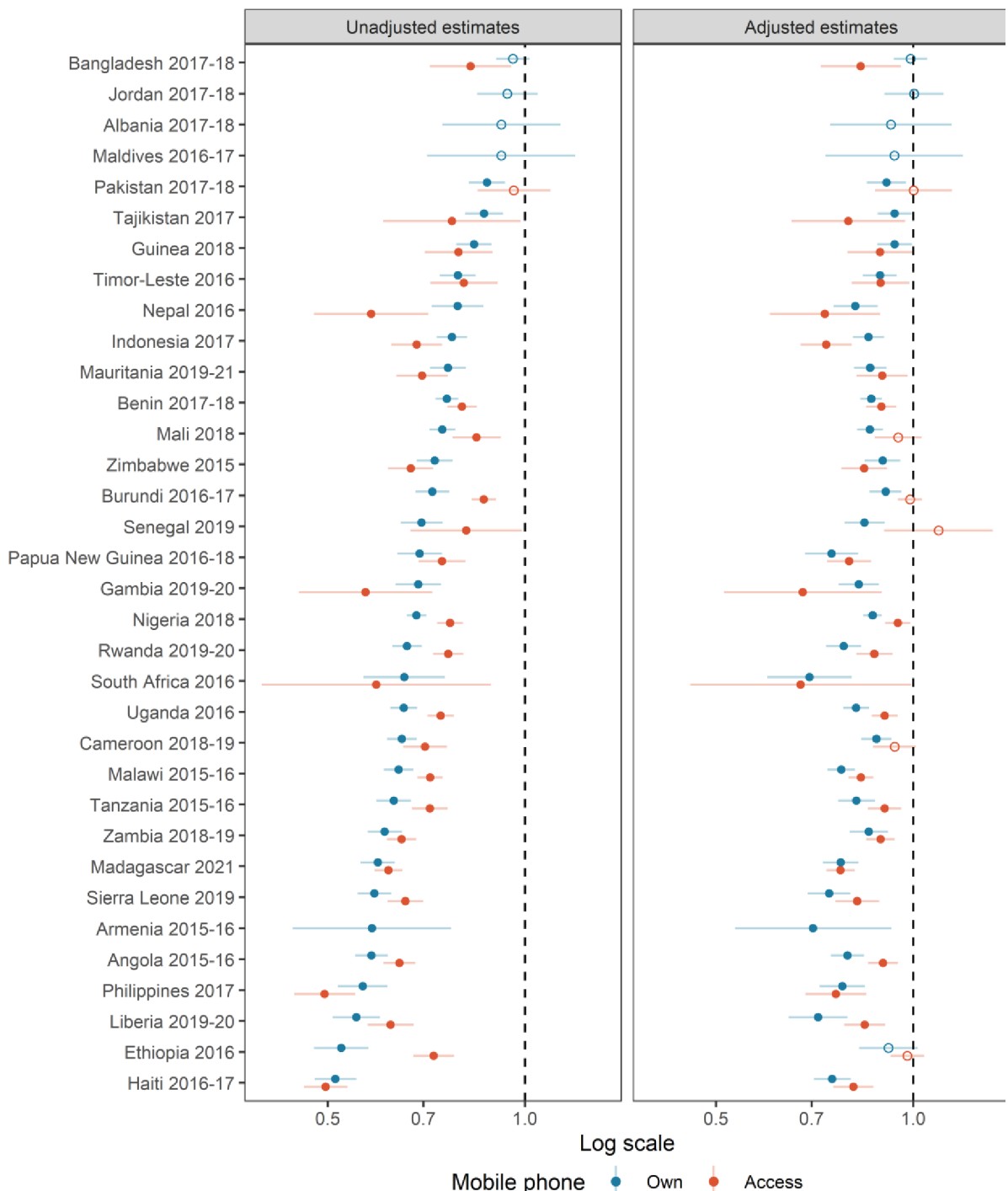

**Figure 2** Unadjusted and adjusted estimates of the effect of owning and having access to a mobile phone on fertility from Poisson regressions. Estimates for access to a mobile phone are not presented for Albania, Armenia, Jordan and Maldives, as access is almost 100%. Fill dots mean that estimates are statistically significant at the 95% confidence level, while empty dots mean that estimates are not statistically significant.

presence of a health professional at delivery, and, to a lesser extent, underweight.

## DISCUSSION

We analysed the social gradient in mobile phone ownership and access in 34 LMICs, and evaluated the implications for fertility and U5M estimates derived from MPS. Consistent with previous research,[16–19] we found that well educated, better-off and urban women are more likely to own and have access to a mobile phone. These differences in the use of mobile phones could bias demographic estimates from MPS.

Fertility and U5M are higher in less advantaged populations,[27 28] groups which are under-represented in MPS. Therefore, MPS are likely to underestimate true levels

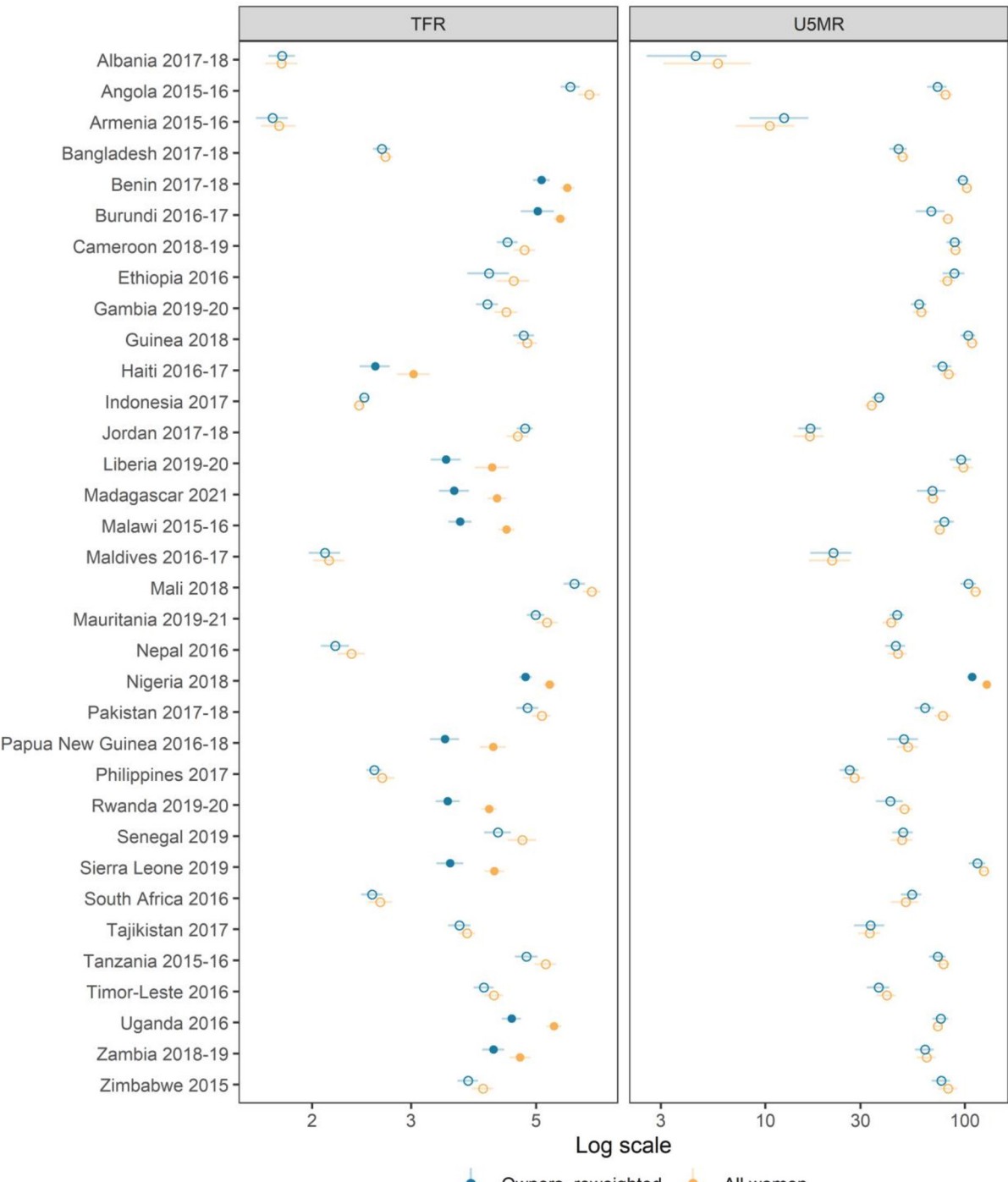

**Figure 3** Under-5 mortality rate (U5MR) and total fertility rate (TFR) estimates testing for poststratification weights from women who own a mobile phone. Fill dots mean that the difference between the actual rate and the reweighted rate is statistically significant at the 95% confidence level, while empty dots mean that the difference is not statistically significant.

of fertility and U5M in the wider population. This is an important first conclusion of this study: MPS will underestimate U5MR and TFR. However, the negative association between U5M and mobile phone ownership disappears once we control for a number of sociodemographic characteristics that can be easily collected in MPS. This is a second, and methodologically, important finding of our study as it suggests that any downward bias in U5M estimates from MPS due to selectivity of respondents can be corrected

using a poststratification procedure. More substantively, our results contradict the hypothesis that at the aggregate level the expansion of mobile phones is associated with lower child mortality as this association seems spurious.[2]

Surprisingly, the negative association between mobile phone ownership and fertility persists in many of the countries included in this study even after controlling for sociodemographic characteristics. From a methodological standpoint, this suggests

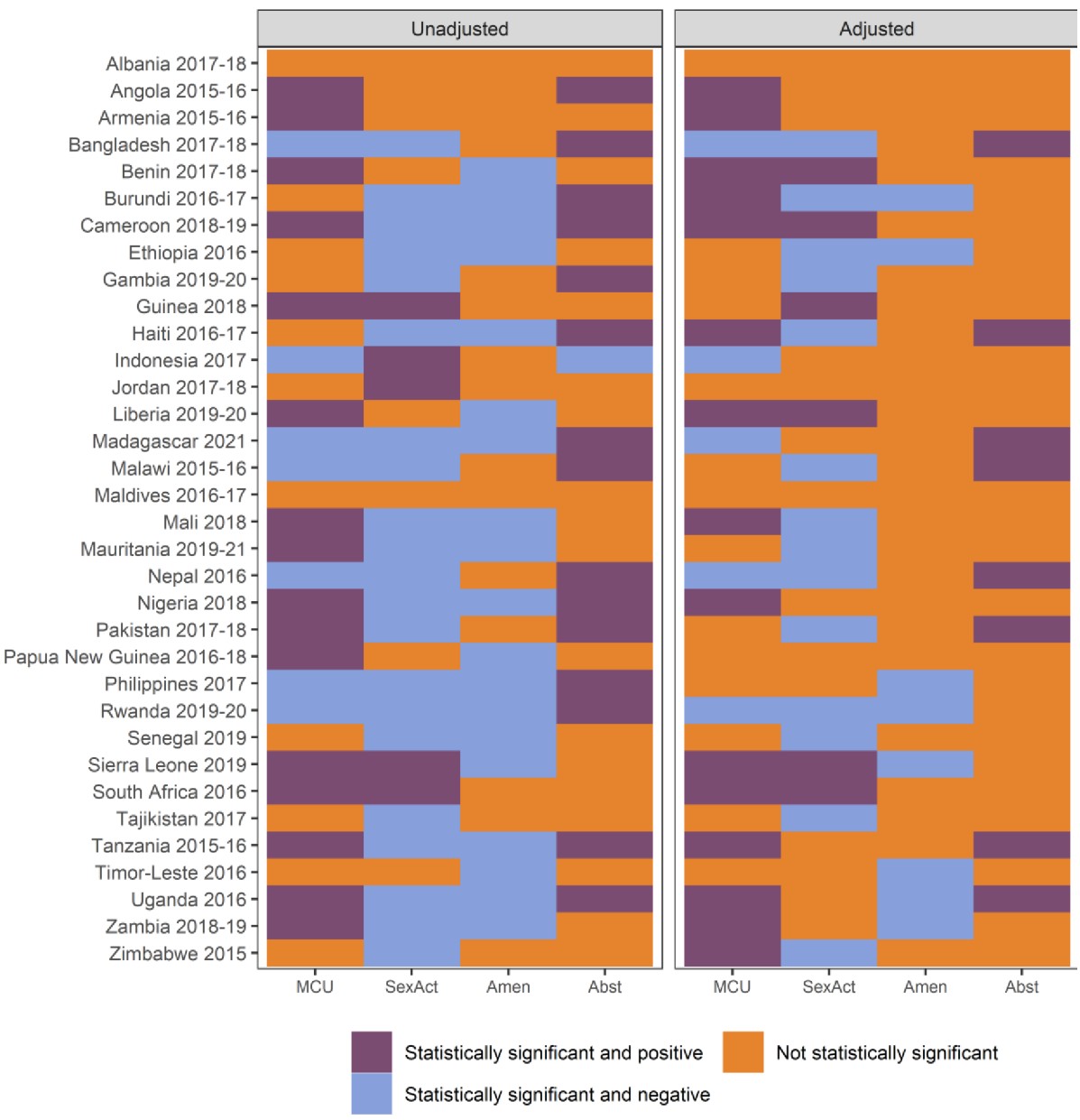

**Figure 4** Estimates of the difference in current use of modern contraceptives (MCU), sexual activity (SexAct), duration of postpartum amenorrhoea (Amen) and duration of postpartum abstinence (Abst), between owners and non-owners of a mobile phone.

that fertility rates could be biased downwards if they are estimated from MPS, even after poststratification. This third important finding is consistent with previous research arguing that mobile phone use is associated with higher rates of contraceptive use.[2 7 20] Our results suggest that the use of modern contraceptives is more likely among owners in two out of five surveys analysed in this study, even after controlling for wealth, education, place of residence, marital status and age. Further, there are some apps that track menstrual periods and send alerts on ovulation days, allowing for better birth control.[2 20] Even less sophisticated models of mobile phones—that is, not smartphones—include calendars that can be used to schedule reminders for tracking the menstrual cycle

or taking contraceptives. In addition, we found that mobile phone owners are less likely to be sexually active in most of the surveys. Higher odds of contraceptive use and lower odds of sexual activity lead to lower fertility rates. It is worth noting that one difference between fertility and mortality is that fertility outcomes can be more easily controlled based on preferences than exposure to illness and injuries. These fertility preferences may be correlated with, but are not fully captured by, the sociodemographic characteristics.

Our results also show that there are no significant differences between owners and non-owners in terms of exclusive breastfeeding. This does not contradict what LeFevre et al[26] found, as they estimated their results using differences in

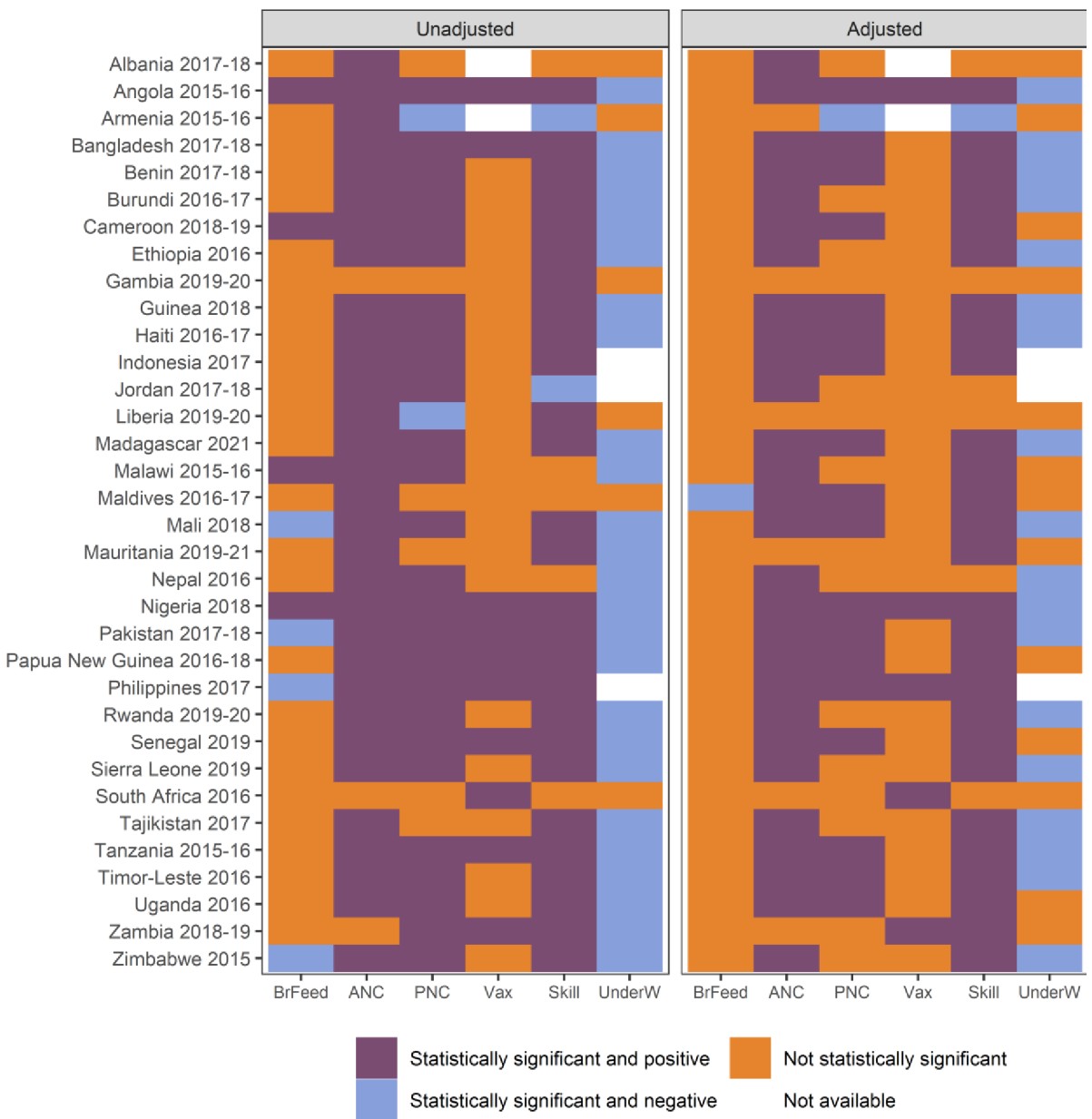

**Figure 5** Estimates of the difference in exclusive breastfeeding (BrFeed), births who had at least four antenatal care visits (ANC), newborns who received at least two signal functions (PNC), children who received basic immunisation (Vax), births delivered by skilled professionals (skill), children who are underweight (UnderW), between owners and non-owners of a mobile phone.

means without controlling for covariates while we have used the association of exclusive breastfeeding conditional on mobile phone ownership and other sociodemographic variables. Our findings can help to explain why the differences in U5MR by ownership disappear when controlling for covariates, since the health benefits from exclusive breastfeeding and basic immunisation could be offsetting the negative effects of no or few prenatal and postnatal checkups or low birth weight.

As for logistical implications for fieldwork, MPS should preferably be conducted at times when household heads are likely to be present so that women in the household who do not own a mobile phone can be surveyed. But biases related to the selection of phone users may remain even after using

this approach. A different way of accounting for selection bias (apart from poststratification), is to make use of sampling quotas (ie, prestratification). This is usually based on a set of simple screening questions like age, rural/urban residence and educational attainment. However, prestratification might be more expensive and it will require greater effort to reach respondents with the desired attributes. Finally, from the perspective of survey designers, conducting complementary face-to-face surveys could be suitable for reaching populations that are typically difficult to cover through MPS. It should be noted that our study evaluates selection bias due to sampling frame limitations (mobile phone access/ownership) and not due to non-response. Non-response is often high in MPS, and considerably higher than in face-to-face

surveys. We did not consider either the recall errors which could be more systematic in MPS due to lower engagement of the respondents.

In conclusion, while MPS are a cheaper alternative or complement to classical face-to-face surveys for estimating demographic and health indicators, estimates derived from MPS cannot be taken at face value owing to the unequal ownership and access of mobile phones in many LMICs. In this study, we have shown that under-5 mortality estimates can be corrected using a poststratification on a number of sociodemographic background characteristics to match nationally representative estimates. Adjusting fertility estimates is less straightforward, most likely because fertility decision-making is an expression of individual preferences in addition to background characteristics and living conditions.

**Contributors** DAS-P and BM led the conception and design of the study. DAS-P and BM analysed the data, produced the estimates and drafted the manuscript. AM-O, OJB and GR reviewed the estimates and helped with the interpretation of the results. All authors critically reviewed the manuscript and approved the final version of the paper. DAS-P is the guarantor of the overall content of this manuscript and had final responsibility for the decision to submit for publication.

**Funding** This study was supported by the Bill & Melinda Gates Foundation grant number INV-023211.

**Disclaimer** The funders of the study had no role in the study design, data analysis, data interpretation, orwriting of the report.

**Competing interests** None declared.

**Patient and public involvement** Patients and/or the public were not involved in the design, or conduct, or reporting, or dissemination plans of this research.

**Patient consent for publication** Not applicable.

**Ethics approval** Not applicable.

**Provenance and peer review** Not commissioned; externally peer reviewed.

**Data availability statement** Data are available in a public, open access repository. The data used in this study are publicly available. It can be found on the DHS Programme website (https://www.dhsprogram.com/data/available-datasets. cfm)

**ORCID iDs**
David A Sánchez-Páez http://orcid.org/0000-0002-7828-8193
Bruno Masquelier http://orcid.org/0000-0002-9585-891X
Ashira Menashe-Oren http://orcid.org/0000-0002-4781-5384
Ololade Julius Baruwa http://orcid.org/0000-0002-0776-528X
Georges Reniers http://orcid.org/0000-0001-6582-1692

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
