## [Reviewer comments · BMJ Open]

ARTICLE DETAILS

TITLE (PROVISIONAL)	Measuring under-five mortality and fertility through mobile phone surveys: an assessment of selection bias in 34 low- and middle-income countries
AUTHORS	Sánchez-Páez, David; Masquelier, Bruno; Menashe-Oren, Ashira; Baruwa, Ololade; Reniers, Georges

VERSION 1 – REVIEW

REVIEWER	Emery, Tom Erasmus University Rotterdam
REVIEW RETURNED	28-Apr-2023

GENERAL COMMENTS	Overall the paper is a wonderful piece of research estimating an important and much understudied issue in the shifting data framework of population policies. I only have one major concern which is that the recommendations focus explicitly on post-stratification which the authors acknowledge is a highly limited form of correction. I understand that they have done this because their recommendations are directed from a researchers perspective. However, their findings may even be useful for survey designers. For them, it could be a sensible strategy to maintain small face-to-face samples for 'hard to reach' populations that are run in parallel to phone surveys. These potential option should be mentioned up front. Simply say that this finding should inform those designing data collections in this area. Adaptive or multi-mode designs would allow for more effective post-hoc calibration whilst also taking advantage of the cost reduction and flexibility of smartphone surveys. This isn't a correction of the authors work per se, but currently the paper reads that post-stratification is the best solution we have and it is not. The best solution would be to design surveys that were ex-ante aware of these potential selection and mode effects. Overall, congratulations on fine work.
--

VERSION 1 – AUTHOR RESPONSE

Reviewer 1

1. I only have one major concern which is that the recommendations focus explicitly on post-stratification which the authors acknowledge is a highly limited form of correction. I understand that they have done this because their recommendations are directed from a researcher's perspective. However, their findings may even be useful for survey designers. For them, it could be a sensible strategy to maintain small face-to-face samples for 'hard to reach' populations that are run in parallel to phone surveys. These potential options should be mentioned up front. Simply say that this finding should inform those designing data collections in this area. Adaptive or multi-mode designs would allow for more effective post-hoc calibration whilst also taking advantage of the cost reduction and flexibility of smartphone surveys.

This isn't a correction of the authors work per se, but currently the paper reads that post-stratification is the best solution we have and it is not. The best solution would be to design surveys that were ex-ante aware of these potential selection and mode effects.

Thank you for raising this point. We agree that our results have implications beyond post-stratification and we have now addressed this issue in two different parts of our manuscript. First, we have mentioned it in the 'Strengths and limitations of this study' section as follows:

"The prevalence of mobile phone surveys is expected to increase in low- and middle-income countries, providing a valuable means of generating comprehensive statistics on demographic and health factors at the national level. However, mobile phone ownership and access will remain unevenly distributed for some time to come, and as this is likely to be correlated with many of the outcomes of interest, it is important to take this into account in the design of the survey. Post-stratification is only one possible solution to calibrate or increase the external validity of the estimates. Other - complementary - solutions include stratified sampling or more complex designs that combine mobile phone interviews with other data collection modalities."

Second, we have added the following sentence (between brackets) to the fifth paragraph of the Discussion section:

"As for logistical implications for fieldwork, MPS should preferably be conducted at times when household heads are likely to be present so that women in the household who do not own a mobile phone can be surveyed. But biases related to the selection of phone users may remain even after using this approach. A different way of accounting for selection bias (apart from post-stratification), is to make use of sampling quotas (i.e., pre-stratification). This is usually based on a set of simple screening questions like age, rural/urban residence and educational attainment. However, pre-stratification might be more expensive and it will require greater effort to reach respondents with the desired attributes. [Finally, from the perspective of survey designers, conducting complementary face-to-face surveys could be suitable for reaching populations that are typically difficult to cover through MPS.] It should be noted that our study evaluates selection bias due to sampling frame limitations (mobile phone access/ownership) and not due to non-response. Non-response is often high in MPS, and considerably higher than in face-to-face surveys. We did not consider either the recall errors which could be more systematic in MPS due to lower engagement of the respondents."